# Inhibitory Effects of Natural Products on Germination, Outgrowth, and Vegetative Growth of *Clostridium perfringens* Spores in Laboratory Medium and Chicken Meat

**DOI:** 10.3390/microorganisms13010072

**Published:** 2025-01-02

**Authors:** Safa Q. Alfattani, Saeed S. Banawas, Mahfuzur R. Sarker

**Affiliations:** 1Department of Microbiology, College of Science, Oregon State University, Corvallis, OR 97331, USA; safa.alfattani@oregonstate.edu; 2Department of Biomedical Sciences, Carlson College of Veterinary Medicine, Oregon State University, Corvallis, OR 97331, USA; 3Department of Biological Science, Collage of Science, Jeddah University, Jeddah 23445, Saudi Arabia; 4Department of Medical Laboratories, College of Applied Sciences, Majmaah University, Al-Majmaah 11952, Saudi Arabia

**Keywords:** *Clostridium perfringens*, natural products, inhibition, spores, essential oil constituents

## Abstract

*Clostridium perfringens* type F is a spore-forming bacterium that causes human illnesses, including food poisoning (FP) and non-foodborne gastrointestinal diseases. In this study, we evaluated the antimicrobial activities of 15 natural products against *C. perfringens* spore growth. Among them, garlic, onion juice, and undiluted essential oil constituents (EOCs) of clove, rosemary, and peppermint showed the strongest activity. Therefore, we examined the inhibitory effects of these products on each stage of the life cycle of *C. perfringens* FP strains, including spore germination, spore outgrowth, and vegetative growth, in laboratory media and chicken meat. Both clove and peppermint oils (at 0.5%; *v*/*v*) inactivated *C. perfringens* spore germination in nutrient-rich trypticase–glucose–yeast extract (TGY) medium. Furthermore, EOCs at (0.1–0.5%) arrested the spore outgrowth of *C. perfringens* in TGY medium. Interestingly, EOCs at 0.5% completely inhibited the vegetative growth of FP isolates during a 6 h incubation in TGY medium. However, even at 4-fold higher concentrations (2%), EOCs were unable to inactivate *C. perfringens* spore growth in contaminated chicken meat stored under abusive conditions. Although some of the natural products inhibited *C. perfringens* spore germination, outgrowth, and vegetative growth in TGY medium, no such inhibitory activity was observed when these products were applied to *C. perfringens* spore-inoculated chicken meat.

## 1. Introduction

*Clostridium perfringens* is a Gram-positive, anaerobic, spore-forming bacteria that can cause intestinal illnesses in humans and animals, making it a significant foodborne pathogen [1,2,3,4]. The pathogenicity of *C. perfringens* is primarily linked to its ability to produce more than 20 toxins [5,6]. This bacterium is divided into seven types (A–G) based on the production of six main toxins [7]. Approximately 5% of type F isolates produce *C. perfringens* enterotoxin, which is responsible for both food poisoning (FP) and non-foodborne (NFB) human gastrointestinal disorders [8]. *C. perfringens* type F is the leading cause of foodborne disease outbreaks in developed countries [9]. Researchers have found that FP and NFB isolates can form metabolically inactive spores that are more resistant to food preservation methods, such as osmosis, extreme pH, high-pressure processing, moist heat, prolonged low-temperature storage, and nitrite, than their vegetative cells [10,11,12]. The surviving spores germinate and multiply in food items, eventually causing toxic infections, diarrhea, and abdominal pain when contaminated food is consumed [13,14]. The risk of contamination increases particularly when food is improperly cooled after cooking or held at unsafe temperatures for extended periods [15,16].

Meat products are the most common vectors of *C. perfringens* transmission. *C. perfringens* cells can proliferate in large numbers in vacuum-packaged meat products because they grow under anaerobic conditions [17]. From 2008 until 2017, at least 24 hospitalizations and 1784 cases of illness were documented in the United States owing to *C. perfringens*-contaminated pig products [18]. Hence, managing *C. perfringens* spores is crucial during packaging following the cooking of meat products [19]. Since *C. perfringens* is regarded as a prevalent harmful bacterial agent in food [9], the food industry aims to create bacterial spore-inactivation solutions that satisfy customer expectations for food safety and quality as substitutes for heat processing technologies or synthetic alternatives.

An innovative alternative to artificial preservatives is using natural products, such as plant extracts and essential oil constituents (EOCs), as food preservatives. These products are Generally Recognized as Safe (GRAS) and can function as an alternate approach for extending the shelf-life of highly perishable food items by limiting the growth of foodborne bacteria [20]. Natural antimicrobial agents have been gaining increased attention from scientific researchers due to their distinctive physicochemical and antibacterial properties [21]. Additionally, since earlier times, natural antimicrobial agents have been widely used for medical and therapeutic purposes, particularly in rural areas of developing countries, to treat health issues such as diarrhea and other gastrointestinal health matters, which Clostridia may cause [22]. Alternative options for replacing synthetic preservatives with natural products have also received greater interest in the past few decades as customers have become increasingly aware of health-related concerns [23] and conventional food preservation methods such as fermenting, brining, and salting, and their growing demand for low-sodium and slightly processed food has grown significantly [23]. Moreover, the addition of natural antimicrobial agents to foods is a cost-effective way to ensure that the active compounds remain in the food and maintain their antimicrobial activity even after opening [24].

EOCs, as natural products, consist of various components, including but not limited to aromatic hydrocarbons, terpenes, terpenoids, alcohols, and acids, and their antimicrobial efficacy is not related to one ingredient or a singular mechanism [25]. Rather, EOCs may act on multiple cellular targets to destroy bacterial cell growth. However, detecting the most effective antibacterial molecules from EOs is sometimes challenging due to their complicated nature [25]. Additionally, the European Commission has approved many natural products, including EOs, as permitted flavor additions in diverse food items [26].

This study’s justification for employing natural plant extracts and EOCs against *C. perfringens* arises from the urgent need to identify alternatives to synthetic antimicrobial agents that have been prohibited in several nations owing to concerns regarding antibacterial resistance [27]. These natural substances have demonstrated significant antibacterial efficacy against *C. perfringens* in both in vitro and in vivo studies [27]. Natural products present a promising option for controlling *C. perfringens* in food items, minimizing issues regarding foodborne diseases, and decreasing dependence on possibly hazardous additives such as nitrite salts [28], sulfites, and chlorides, which are used to prevent the proliferation of foodborne microorganisms, as these chemicals have been linked to carcinogenic consequences and many health concerns [23].

While significant research has been conducted on using natural substances as antibacterial agents in meat products, evidence of their effectiveness against *C. perfringens* is limited. This study aimed to (1) screen a variety of natural products as antibacterial agents to inhibit the growth of *C. perfringens* spores, (2) assess the inhibitory effects of those products that show the highest effects on the spore germination, outgrowth, and vegetative growth of type F *C. perfringens*, and (3) evaluate the potential application of selected natural products as antibacterial agents to prevent the growth of spores in *C. perfringens*-inoculated chicken meat preserved under harsh conditions.

## 2. Materials and Methods

### 2.1. Bacterial Strains and Growth Conditions

This study’s pathogenic *C. perfringens* type F isolates comprise three FP strains (SM101, NCTC10239, and E13). The sources of these bacterial strains have been described in detail previously [29,30,31]. *C. perfringens* isolates were stored in stock cultures at –20 °C using a cooked meat medium (Difco, BD Diagnostic Systems, Sparks, MD, USA). To revive bacterial growth, 0.1 mL of cooked meat culture was inoculated into 10 mL fluid thioglycollate medium (FTG) (Difco, BD Diagnostic Systems, Sparks, MD, USA) and cultured overnight at 37 °C. TGY broth (3% trypticase, 2% glucose, 1% yeast extract, and 0.1% L-cysteine) was used to test the vegetative growth and spore outgrowth of *C. perfringens* isolates [14].

### 2.2. Spore Preparation and Purification

*C. perfringens* spores were prepared and purified as previously described [14]. In brief, approximately 0.1 mL of each *C. perfringens* stock culture was transferred to 10 mL FTG and incubated overnight, then 0.4 mL from the previous culture was transferred into another FTG and grown for 9–12 h at 37 °C. Next, 0.4 mL from the second FTG was inoculated into 10 mL Duncan–Strong (DS) sporulation medium (1.5% proteose peptone, 0.4% yeast extract, 0.1% sodium thioglycolate, 0.5% sodium phosphate dibasic [Na_2_HPO_4_] [anhydrous], and 0.4% soluble starch) [32], and incubated for 18–24 h at 37 °C. After 24 h of incubation, sporulation was examined using a phase-contrast microscope (Leica MDLS, Leica Microsystems, Wetzlar, Germany). Scaling up the aforementioned technique resulted in a large volume of spores. Spores were purified by repeatedly washing and centrifuging in cold, sterile distilled water (SDW) until they were >99% free of sporulating cells, cell debris, and germinated spores, as assessed by phase-contrast microscopy. Purified spores were suspended in SDW to achieve an optical density at 600 nm (OD_600_) of approximately 6 using a SmartSpec^TM^ 3000 spectrophotometer (Bio-Rad Laboratories, Hercules, CA, USA) and stored at –80 °C until use.

### 2.3. Preparation of Antibacterial Solutions

The natural substances used in this study were evaluated based on their historical usage, contemporary popularity, and practicality [33]. The products were classified into two main categories: a raw group including garlic juice, onion juice, ginger juice, garlic powder, cinnamon powder, turmeric powder, and ginger powder, and processed health supplements comprising garlic tablets, ginger tablets, turmeric capsules, cinnamon capsules, coconut oil, peppermint oil, clove oil, and rosemary oil. These substances were purchased from stores, pharmacies, and healthy food suppliers in Corvallis, OR, United States. As mentioned previously [33], products such as garlic bulbs, ginger, and onions were rinsed with SDW and then crushed with a mortar and pestle or grinder. The juices from these products were extracted by filtering through cheesecloth. Other products, in the form of dry powdered spices or tablets, were suspended in either SDW or 20% dimethyl sulfoxide (DMSO) and gently agitated at room temperature 22 °C overnight. The spice suspensions were centrifuged, and the supernatants were stored for testing. For the remaining products, including essential oils, 50% (*v*/*v*) EOC stock solutions were prepared, and further diluted as required.

### 2.4. Screening Natural Product Activity Against C. perfringens

All natural products were screened for inhibitory activity against FP *C. perfringens* strains using an agar well diffusion assay. To revive bacteria, the cooked meat stock culture was inoculated into FTG medium and incubated at 37 °C for 18–24 h, and 0.4 mL of the previous culture was added to 10 mL fresh TGY broth and incubated for 3 h at 37 °C; the TGY-grown culture was used to conduct the agar well diffusion assay as previously described [33]. In summary, *C. perfringens* TGY-grown cultures were distributed on brain heart infusion (BHI) agar, with 8 mm wells cut into each tested plate. Aliquots (100 µL) of the natural products under study were pipetted into each well, and the same volume of SDW or 20% DMSO was added to individual wells as negative controls. Plates were incubated at 37 °C for 24 h in an anaerobic chamber, and the growth of *C. perfringens* in the surroundings of the well was evaluated. The absence of bacterial growth around the substance-containing wells indicated that the product was an antibacterial agent against the *C. perfringens* isolate tested. Growth-inhibition zones were estimated to the nearest millimeter using a transparent ruler.

### 2.5. C. perfringens Spore Germination in the Presence of Natural Products

A *C. perfringens* spore germination assay was performed as previously described [14]. Briefly, spore suspensions at OD_600_ ≈ 6 were heat-activated at 80 °C for 10 min, cooled at room temperature 22 °C for 5 min in a water bath, and kept at 37 °C for 10 min. Suspensions were then inoculated into TGY alone or supplemented with different concentrations of antibacterial agents (0.1–0.5%) in a 96-well microtiter plate. To track spore germination, the OD_600_ of the spore cultures was measured using a Synergy^TM^ MX multi-mode microplate reader (BioTek^®^ Instruments, Inc., Winooski, VT, USA). With complete spore germination, OD_600_ decreases by approximately 60% [14]. Phase-contrast microscopy further confirmed the spore germination rate, as germinated spores turn to phase dark spores. The germination rate inhibition was determined as the percentage loss of OD_600_ compared to the original value after 60 min of incubation in TGY with or without the tested antimicrobials using the following equation [34]:Germination rate % = (OD_600_ decrease in treated cells/OD_600_ decrease in control cells) × 100

All spore germination experiments were repeated at least twice, with two independent spore preparations for each isolate.

### 2.6. C. perfringens Spore Outgrowth in the Presence of Natural Products

The capacity of the spores to outgrow was investigated in a TGY vegetative growth medium. First, spore suspensions (200 µL) at OD_600_ ≈ 6.0 were heat-activated at 80 °C for 10 min then inoculated into 10 mL pre-heated TGY broth alone or supplemented with different concentrations of natural products. Cultures were incubated at 37 °C, and bacterial growth was observed by measuring OD_600_ at 60 min intervals for up to 180 min. Spore outgrowth findings were expressed as a percentage increase in OD_600_, and the inhibition of spore outgrowth was determined as follows [34]:Spore outgrowth % = (OD_600_ increase in treated cells/OD_600_ increase in control cells) × 100

All the experiments were conducted at least twice using different spore preparations.

### 2.7. C. perfringens Vegetative Growth in the Presence of Natural Products

Aliquots (~0.1 mL) of the cooked meat stock cultures were transferred to 10 mL FTG and incubated at 37 °C for 18–24 h. Subsequently, 0.4 mL from each culture was transferred to 10 mL TGY and grown at 37 °C for 3 h. Next, 0.4 mL aliquots of the previous cultures were inoculated into 10 mL TGY alone (control) or supplemented with different concentrations of antimicrobials. Vegetative growth was determined by measuring the OD_600_ at different times up to 24 h. OD_600_ results from liquid cultures were compared with the number of colony-forming units per mL (CFU/mL) from each culture. Briefly, samples from each treatment were collected at 0 and 6 h post-inoculation, serially diluted, and plated onto BHI agar (Difco). Plates were incubated anaerobically at 37 °C for 18–24 h and bacterial colonies were counted. All experiments were conducted separately, at least in duplicate, and the values were reported as means.

### 2.8. C. perfringens Spore Growth in Cooked Meat in the Presence of Natural Products

The antibacterial activity of the selected natural products against *C. perfringens* spore growth in cooked chicken meat was evaluated as previously described [35]. Briefly, meat samples were obtained from a local store in Corvallis, OR, USA, and prepared as previously described [36]. An amount of 10 g/bag of ground chicken meat samples was autoclaved and stored at −20 °C until use. Suitable amounts (0.5, 1.0, and 2.0%; *v*/*v*) of natural product stock solutions were added to the ground chicken. Then, 0.1 mL aliquots of a spore mixture containing three spore isolates at a concentration of ~10^8^ spores/mL were manually mixed with the meat. Each experimental replicate included a negative control to verify that the meat samples were free of naturally occurring microorganisms. Ground chicken meat sample mixtures (including the meat, spore cocktail, and natural [putatively antibacterial] product) were cooked at 80 °C for 20 min and cooled in a water bath at room temperature for 20 min. Two sample bags were used for each treatment, one was used to determine the initial population of *C. perfringens* in chicken meat, whereas the other was transferred aseptically to a sterile petri dish and held under anaerobic conditions at room temperature for 6 h. To evaluate the populations of *C. perfringens*, each meat sample was placed into a stomacher bag, combined with 90 mL 0.1% (*w*/*v*) peptone water, serially diluted, and plated onto BHI agar plates that were incubated anaerobically at 37 °C for 24 h. The colonies were counted. The results were expressed as CFU/g [35].

### 2.9. Statistical Analysis

One-way analysis of variance (ANOVA) and nonparametric approaches were employed to determine the statistical significance between more than two data sets, followed by Tukey’s multiple comparison post-test if the *p* value was significant, utilizing Prism version 10.0 (GraphPad, Boston, MA, USA). A Student’s two-tailed t-test was applied to determine whether a significant difference existed between the two data sets. *p* values of <0.05 were considered significant. Mean values were computed, and the error bars in all figures denote the standard deviation from the mean. All experiments were independently performed at least in duplicate.

## 3. Results and Discussion

### 3.1. Screening Natural Product Activity Against C. perfringens

An agar well diffusion assay was performed to test the activity of 15 natural compounds at high concentrations against *C. perfringens* SM101 (Table 1 and Table 2). In total, twelve products displayed inhibition zones of several sizes. Among the seven raw substances screened, 100% *v/v* garlic juice was the most effective antibacterial agent, exhibiting the largest zone of inhibition (29.6 mm), as shown in Table 1. Additionally, all the investigated processed compounds, except for ginger tablets and coconut oil, showed antibacterial activity against *C. perfringens*, with garlic tablets being the most effective (Table 2). Similar data were obtained in former studies with many Gram-positive bacteria including *C. perfringens* strain ACTC13124, different isolates of *C. difficile*, and *Staphylococcus aureus* ATCC29213 [33]. The inhibitory effect of garlic against *C. perfringens* could be related to the primary bioactive component in garlic called allicin which accounts for its antibacterial properties [37]. Additional components in garlic, including saponins and flavonoids, may also enhance its antibacterial properties [37]. Given that all garlic forms demonstrated excellent solubility in both SDW and DMSO, they provide a practical agent for investigation.

Importantly, each experiment included negative controls in which 20% DMSO or coconut oil (used as solvents for some essential oils) were added to the agar wells, and no inhibition zones were observed. Overall, natural compound preparations dissolved better in 20% DMSO than in SDW and resulted in larger inhibition zones. For example, both turmeric powder and capsules showed no inhibitory zones when SDW was used as the solvent. Similarly, 20% *w/v* of cinnamon powder and capsules showed negligible activity when dissolved in SDW. In contrast, when dissolved in 20% DMSO, both presentation forms of cinnamon and turmeric produced inhibition zones, ranging between 11.75 and 13.4 mm and between 9.1 and 11 mm, respectively. An increase in the solubility of the products in the agar owing to DMSO could explain the expansion of the inhibition zones. Our results are in agreement with those of previous reports using 20% DMSO and SDW as solvents of various formulae of natural products [33]. Specifically, when both cinnamon and garlic in dissimilar forms were dissolved in 20% DMSO, larger inhibition zones against *C. perfringens* ATCC13124 were detected (11.6–16.1 mm and 15.5–26.5 mm, respectively), compared with those formed when SDW was used as a solvent. However, it is important to understand that the agar well diffusion assay is a rapid and easy method to obtain and analyze data [38,39], and parameters, such as product diffusion, medium composition, and incubation conditions, may affect the extent of the inhibitory zone and the susceptibility of the microorganism under study [38,40]. Although not as sensitive as other antibacterial activity analyses, this method is commonly used for rapid screening and enables the evaluation of results obtained using different methods [40]. Therefore, from the two groups of tested natural substances (raw and processed), we selected the products that showed the larger inhibition zones for further investigation against *C. perfringens* FP strains. This included garlic juice (the most effective among the garlic forms examined), onion juice, and EOCs, including those from clove, rosemary, and peppermint.

### 3.2. Inactivation Activity of Natural Products Against C. perfringens Spore Germination

Given that the spores of *C. perfringens* germinate before evolving into growing vegetative cells, the inhibitory effect of the selected natural products on spore germination was initially investigated. To determine the natural product concentrations that could inhibit spore germination, we evaluated the extent of germination of the FP strain SM101 in TGY alone or supplemented with different concentrations of inhibitors, ranging from 0.1 to 0.5%. *C. perfringens* SM101 showed complete spore germination after 60 min of incubation at 37 °C when inoculated in TGY broth medium alone (Figure 1). However, when inoculated in TGY supplemented with the selected natural products, SM101 spores did not germinate in the presence of 0.5% clove and peppermint oils and even germinated poorly in the presence of 0.25% and 0.1% peppermint oil. These findings revealed that clove and peppermint oils exhibited mild and strong inactivation effects, respectively, on SM101 spore germination. In agreement with a previous study, spores of *Bacillus subtilis* generated a reduction in the extent of germination relative to control spores in the presence of clove, which indicates the inhibitory activity of this substance against spore germination [41].

Conversely, the spores of SM101 successfully germinated in TGY alone or supplemented with 0.1–0.5% garlic and onion juices, 0.1–0.25% clove oil, and all tested concentrations of rosemary oil, demonstrating that garlic juice, onion juice, clove oil at concentrations below 0.5%, and rosemary oil do not affect spore germination (Figure 1A, B, D, and E, respectively). In another study on *B. cereus* T, the EO of rosemary at a specific concentration was partially sporicidal [42].

The natural products’ inhibitory effects on spore germination were further confirmed using two additional FP strains of *C. perfringens* type F, E13, and NCTC10239, which yielded almost identical results. Overall, our findings showed that the extent of inhibition of spore germination varied significantly among the selected natural substances and was significantly influenced by the type and concentration of these products.

### 3.3. Inhibitory Effect of Natural Products Against C. perfringens Spore Outgrowth

We assumed that some selected natural products, such as garlic juice, onion juice, and rosemary oil, that were not effective against spore germination could work efficiently during the period of spore outgrowth, as this phase of spore growth is more sensitive to antimicrobials than that of spore germination. Additionally, *C. perfringens* outgrowth may require lower concentrations of EOCs to prevent spore germination in the TGY medium. In the event of a lack of all tested natural products, spores of all examined strains (SM101, E13, and NCTC10239) reached the outgrowth step after approximately 180 min of inoculation into TGY medium and incubation at 37 °C (Figure 2A–E). Nevertheless, the presence of any of the concentrations (0.1–0.5%; *v*/*v*) of the EOCs (clove, rosemary, and peppermint) under study was enough to significantly block spore outgrowth in all tested FP isolates (*p* < 0.05) as compared with that of the control in TGY. Only a slight outgrowth of the spores of isolate E13 was detected after approximately 180 min of inoculation with 0.1% rosemary oil. These results are similar to an earlier study in which the spores of *B. subtilis* did not outgrow when the spore culture was stored with different concentrations of clove, demonstrating its inactivation activity as sporostatic at certain doses [43].

In contrast, when a high concentration (1%, *v*/*v*) of garlic and onion juices was added to TGY, FP strain spores were able to outgrow for approximately 120 min at 37 °C, indicating that garlic and onion juices were not sufficient to arrest *C. perfringens* outgrowth even at high concentrations (Figure 2D,E). In summary, our results revealed that clove, rosemary, and peppermint oils in the TGY medium are potent blockers of the spore outgrowth of the FP isolates tested.

### 3.4. Inhibitory Effect of Natural Products Against C. perfringens Vegetative Cell Growth

The vegetative cell growth of the representative FP isolate SM101 was studied in TGY in the presence of selected natural products at various concentrations (0.1–0.5%). Interestingly, the results demonstrated that all tested EOCs at 0.5% were capable of completely inhibiting vegetative cell growth over a 24 h time frame (Figure 3). Moreover, decreasing EOC concentrations to 0.25% or 0.1% similarly caused a delay in bacterial cell growth relative to that of the control sample during a 24 h incubation period in TGY (Figure 3). We then expanded our tests to include FP strains E13 and NCTC10239 to assess whether the inhibitory activity of EOCs on the vegetative growth of pathogenic *C. perfringens* was strain-dependent. The growth-inhibition patterns of the other two tested FP isolates were similar to that of SM101, as 0.5% EOCs arrested vegetative growth for 24 h compared with the control. However, a slight variation in the growth curves of both E13 and NCTC10239 was observed when TGY was supplemented with other percentages of clove, rosemary, and peppermint oils. Furthermore, when we examined the effect of a higher concentration of garlic and onion juice (1%) against FP SM101, we found that this isolate was resistant to both substances (Figure 3D, E). Similarly, the vegetative cell growth of E13 and NCTC10239 in TGY supplemented with 1% garlic and onion juices was not inhibited.

Among the five natural products tested (three EOCs and two juices), only clove, rosemary, and peppermint oils were potent inhibitors of the vegetative cell growth of the FP strains SM101, E13, and NCTC10239.

The antibacterial effect of EOC ingredients is primarily attributed to their hydrophobic nature, enabling them to penetrate bacterial membranes, damage their integrity, and cause the release of cellular components. The cell wall construction of Gram-positive bacteria enables the passage of hydrophobic compounds, impacting both the cell wall and intracellular constituents [44]. As noticed from the above findings, the addition of EOCs at different concentrations to the TGY medium showed different inhibition activities against *C. perfringens* FP strains. However, clove and peppermint oils were among the products that demonstrated their antibacterial effects against spore germination, spore outgrowth, and vegetative cell growth in a laboratory medium.

The antibacterial activities of these substances against *C. perfringens* are mostly attributed to the major bioactive compounds in clove and peppermint oils. Eugenol is a significant bioactive component in clove essential oil, exhibiting diverse biological functions including antibacterial effects [45]. The antibacterial properties of eugenol have been extensively studied. Eugenol may increase bacterial cell membrane permeability, causing intracellular leakage and cell death [46]. Peppermint essential oil has a greater quantity of bioactive components. Menthol is a main component of peppermint essential oil, which demonstrates its antibacterial properties against many foodborne pathogens [47]. However, the antibacterial mechanisms of eugenol and menthol against *C. perfringens*, especially FP strains, need to be investigated.

The findings of the inhibitory effect of clove oil against *C. perfringens* FP strains obtained from this study are supported by previous studies [48] that proved that clove oil has antibacterial activity against other pathogens including *S. aureus*, *Listeria monocytogenes*, and *Escherichia coli* with efficacy relying upon its concentration and the duration of exposure [49,50,51]. Further, the results of the inactivation activity of peppermint oil against *C. perfringens* FP isolates are similar to earlier investigations that have demonstrated its efficacy against *B. cereus*, *S. aureus*, and *Enterococcus faecalis* [52].

### 3.5. Inhibitory Effects of Natural Products on C. perfringens Spore Growth in Chicken Meat

To verify our results in vivo, and to assess the actual implementation of selected natural substances that showed inactivation activity against *C. perfringens* spore germination or outgrowth, or vegetative cell growth in foods that are vulnerable to contamination by *C. perfringens*, we examined the impact of EOCs, including clove, rosemary, and peppermint oils, on a meat model maintained in harsh conditions, as described earlier [35,36]. Furthermore, since FP *C. perfringens* type F isolates cause most disease outbreaks [53], a mixture of three FP strains (SM101, E13, and NCTC10239) was used to assess the inhibitory effects of EOCs on spore growth in chicken meat. Notably, those EOC doses (0.5%) that suppressed *C. perfringens* growth in TGY in the laboratory were insufficient to control spore germination and outgrowth in cooked chicken meat after 6 h of storage in anaerobic environments; that is, approximately, 2.5 log increases in *C. perfringens* abundance were observed in chicken meat after incubating the samples under extreme conditions for 6 h, regardless of the EOC tested (Figure 4). Moreover, chicken samples with 1% and 2% concentrations of the tested natural substances presented approximately a 2 log CFU/g increase in *C. perfringens* viable cell counts after 6 h of incubation at room temperature, similar to those observed in control samples without any antimicrobial agent. Our results show that, even at a high concentration (2%), the tested EOCs did not inhibit the growth of a cocktail of FP spores of *C. perfringens* isolates in the tested meat product.

It is important to mention that the chicken meat samples tested in this study were kept in severely abused situations to simulate the worst-case scenario, such as the one that might occur if samples were kept at a warm temperature, promoting the growth of *C. perfringens* before meat ingestion.

In a previous study, ground beef samples containing 10% clove oil, a high concentration compared with the percentages used in this investigation, completely inactivated the growth of *L. monocytogenes* during three days, irrespective of storage temperature [54]. Moreover, earlier reports showed that the final counts of lactic acid bacteria (LAB) cell growth were influenced by the incorporation of different concentrations of rosemary oil with minced meat [55]. In addition, peppermint oil consists of many compounds, with menthol being the most bioactive component mentioned above. A concentration of 2% peppermint oil can suppress 2 and 3 logs of *L. monocytogenes* and *Salmonella typhimurium* growth in minced meat, respectively [47]. The inhibitory effect of these agents against *L. monocytogenes*, LAB, and *S. typhimurium* could be explained by the fact that this bacterial genus is non-spore-forming, making it susceptible to inactivation by mild inactivation treatment.

The lack of antibacterial activity of the tested substances against *C. perfringens* in meat could be attributed to the fact that spores and vegetative cells of FP isolates are resistant to various food preservation procedures including EOCs [11,12]. Moreover, the efficacy of these products in complex food matrices such as meat products is frequently impacted by their limited water solubility, and the intrinsic characteristics of the meat, including fat, protein, water content, pH, and salt concentration, which in turn affects interactions with these components [56], therefore weakening the EOCs’ inhibitory effects [57]. High protein and fat and reduced water content may decrease the action of some EOs in meat products [58].

In vitro, studies provide positive outcomes from applying EOCs as natural preservatives owing to their antibacterial properties. In food systems, the bioactivity and efficacy of these products are often reduced, requiring doses higher than those determined in vitro which develop undesirable sensory effects [58]. However, this issue could be overcome by incorporating EOCs with other food preservation approaches, such as temperature, irradiation, and pulsed electric fields, to decrease the dosage of oils required for pathogen inactivation, or by assessing an effective synergistic combination with other oils. Numerous essential oils had inhibitory effects on the vegetative growth and spore germination of Bacillus species and *Geobacillus stearothermophilus* when combined with other oils [59]. Such mixtures enhanced both bacteriostatic and bactericidal properties, demonstrating additive or synergistic effects [59]. These findings indicate that essential oils and their combinations may effectively manage spore-forming bacteria in food preservation and other contexts. This calls for more study into their potential as natural antibacterial agents. Moreover, an option to reduce the interaction of EOCs with food elements including fats and proteins is to encapsulate the oil in a suitable biodegradable substance such as chitosan, which may provide regulated release while maintaining its biological effect.

## 4. Conclusions

The current study showed that a collection of natural products can inhibit different growth forms in the life cycle of *C. perfringens* FP isolates including spore germination, spore outgrowth, and vegetative cell growth in rich laboratory media. Clove and peppermint oils at 0.5% showed the most potent action against *C. perfringens* spore germination. Further, the three tested EOCs at the lowest concentration (0.1%) suppressed the spore outgrowth of *C. perfringens*. Interestingly, 0.5% of EOCs blocked the vegetative cell growth of FP isolates during 6 h incubation in a TGY medium. Considering these products’ efficiency under laboratory conditions, these substances, even at 4-fold higher concentrations (2%), were unable to inhibit *C. perfringens* spore growth in complex food matrices such as chicken meat that had been stored improperly for an extended period.

The evolution from in vitro investigations to in vivo studies for assessing the effectiveness of EOCs has consistently presented additional challenges. Hence, future research should focus on a comprehensive survey of EOCs’ antibacterial capabilities, target species, their mechanisms of action, and the pathways influenced by the constituents of EOs. This information will offer knowledge and reveal novel perspectives about available applications in the food sector to facilitate the usage of EOCs as food preservatives and prevent the spread of foodborne diseases. To conclude, the findings of this study illustrated the efficacy of natural products that have been evaluated for their application as food preservatives. Nonetheless, their use in food items has been limited on a commercial food basis due to the need for higher dosages to achieve effective antibacterial activity, and the optimization of the quantity, source, and active component of these products, specifically EOCs for food applications, remains underdeveloped. Hence, surveillance should be exerted if these natural products are added to meat products to decrease or eradicate bacterial spores, particularly the spore growth of FP *C. perfringens*.

## Figures and Tables

**Figure 1 microorganisms-13-00072-f001:**
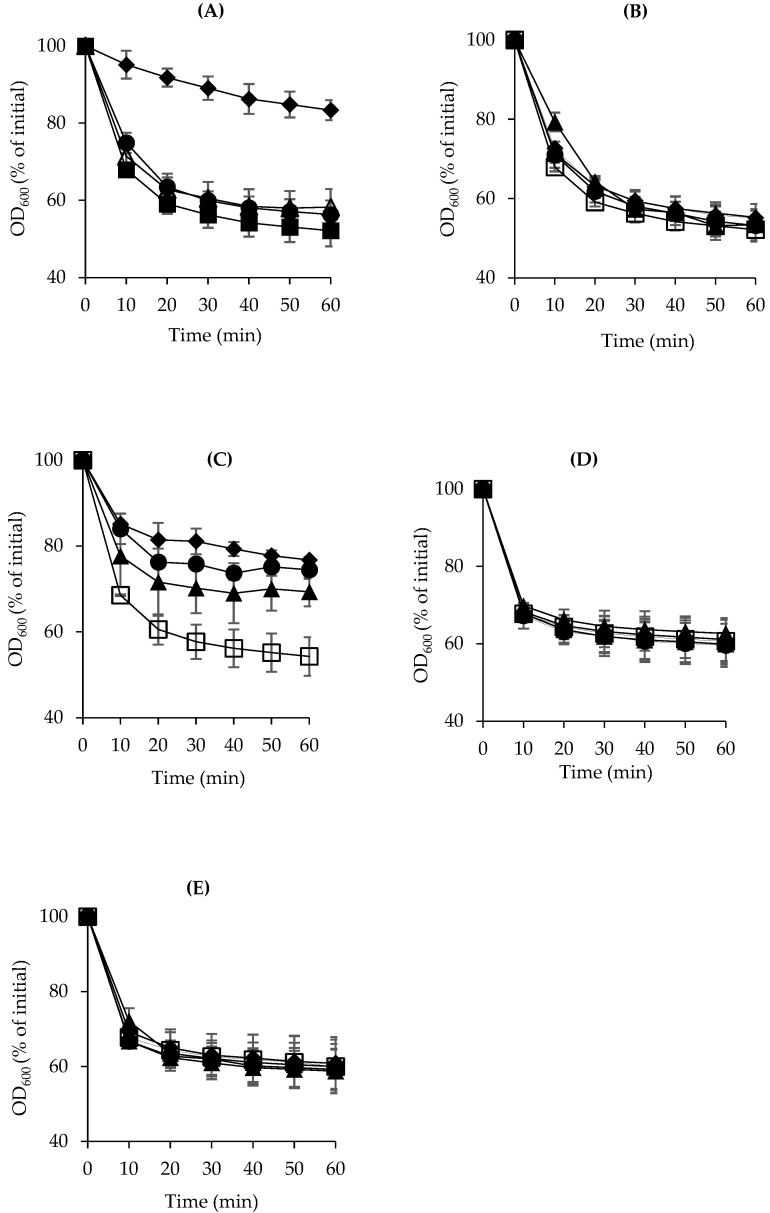
Inhibitory effects of selected natural products on the germination of *C. perfringens* SM101 spores. SM101 spores were heat-activated and inoculated in TGY broth alone (□) or supplemented with 0.5% (♦), 0.25% (●), or 0.125% (▲) of (**A**) clove oil, (**B**) rosemary oil, (**C**) peppermint oil, (**D**) garlic juice, or (**E**) onion juice. The percentage of germination was monitored by measuring the decrease in OD_600_ over time. Note that, in panel (**A**), data symbolized by (●) and (▲) are overlapped; in panel (**B**), data symbolized by (□), (♦), (●), and (▲) are overlapped; and, in panels (**D**,**E**), data symbolized by (□), (♦), (●), and (▲) are overlapped. Values represent the average from duplicate experiments using two different spore preparations. Error bars represent standard deviations from the means.

**Figure 2 microorganisms-13-00072-f002:**
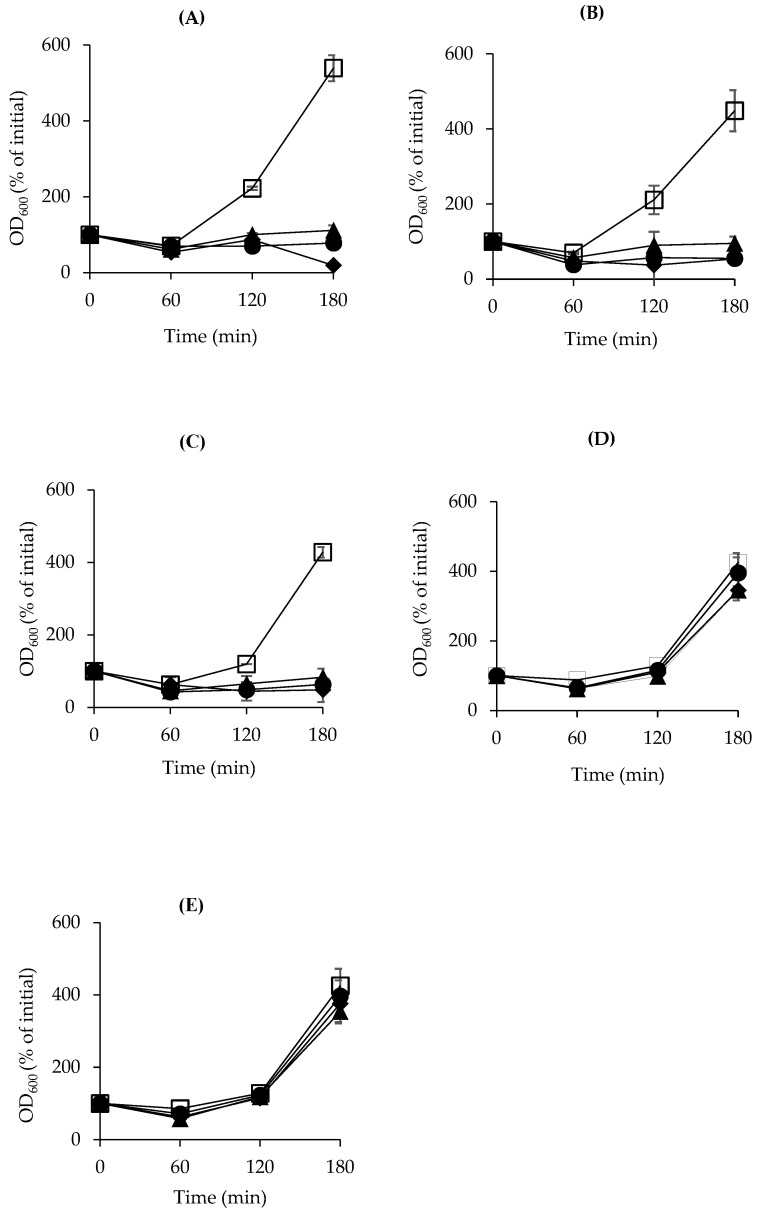
Inhibitory effects of natural products on spore outgrowth of strain SM101. SM101 spores were heat-activated and inoculated into TGY broth alone (■) or supplemented with 0.5% (♦), 0.25% (●), or 0.1% (▲) of (**A**) clove oil, (**B**) rosemary oil, (**C**) peppermint oil, (**D**) garlic juice, or (**E**) onion juice. Spore outgrowth was monitored hourly for up to 3 h of incubation at 37 °C by measuring OD_600_. Note that, in panels (**A**–**C**), data represented by (♦), (●), and (▲) are overlapped; whereas, in panels (**D**,**E**), data represented by (□), (♦), (●), and (▲) are overlapped. Each data point represents the average of duplicate experiments using two different spore preparations. Error bars represent standard deviations from the means.

**Figure 3 microorganisms-13-00072-f003:**
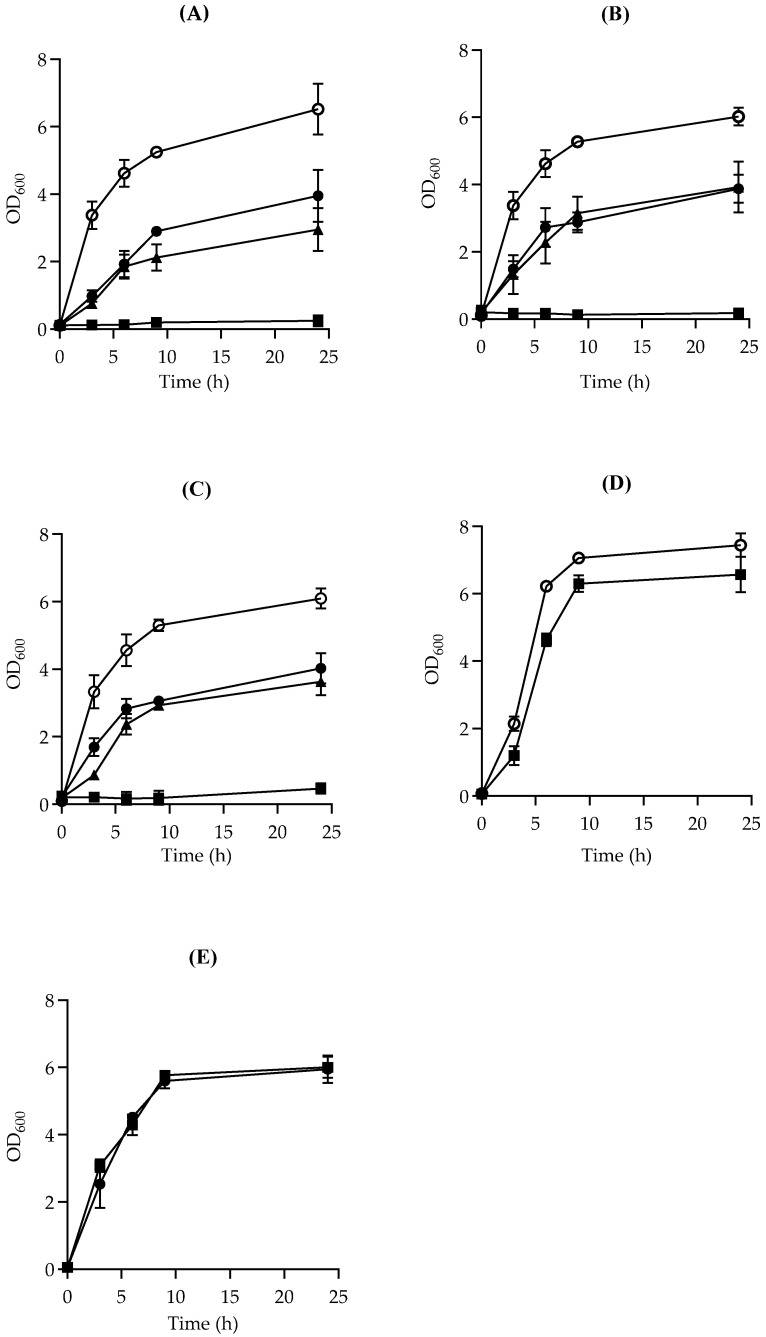
Effects of natural products on the vegetative cell growth of *C. perfringens* strain SM101. Vegetatively growing (3 h TGY-grown culture) SM101 cells were inoculated into fresh TGY medium containing different concentrations of (**A**) clove oil, (**B**) rosemary oil, (**C**) peppermint oil, (**D**) garlic juice, or (**E**) onion juice. Growth was determined by measuring OD_600_ at regular time intervals. Natural product concentrations are represented by (○), 0%; (●), 0.1%; (▲), 0.25%; and (■), 0.5%. Note that, in panels (**A**,**B**), data represented by (●) and (▲) are overlapped, and, in panel (**E**), data represented by (●) and (■) are overlapped. Error bars represent the standard deviations from the means of at least two experiments carried out independently.

**Figure 4 microorganisms-13-00072-f004:**
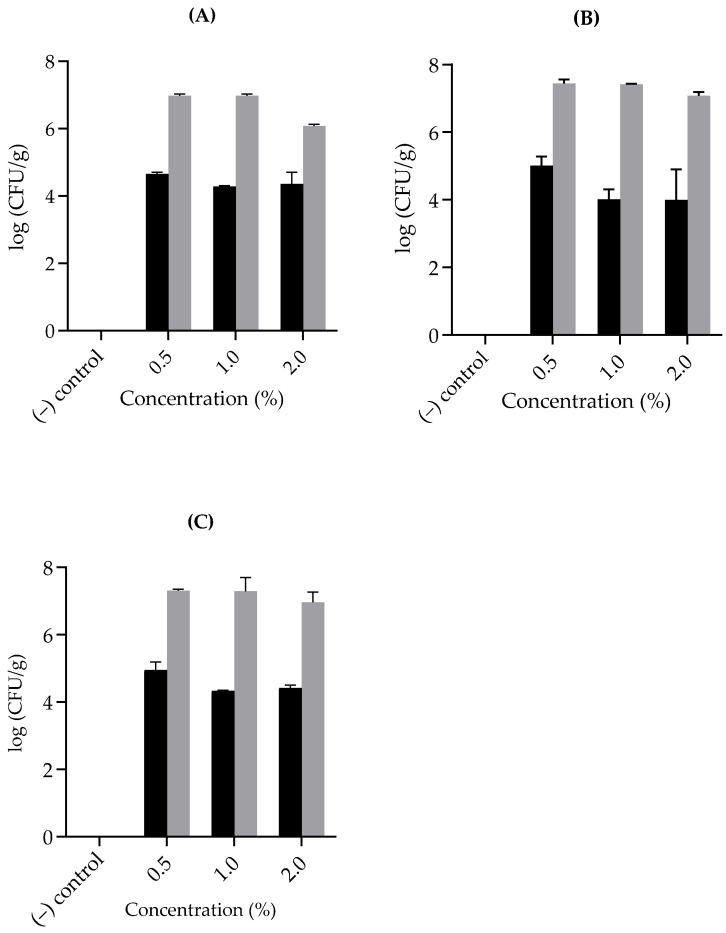
Inhibitory effects of natural products on *C. perfringens* SM101 spore growth in cooked chicken meat. A mixture of spores from three strains of *C. perfringens* was inoculated into cooked chicken samples containing diverse concentrations (% *v*/*v*) of (**A**) clove, (**B**) rosemary, and (**C**) peppermint oils. The number of colony-forming units (CFUs) formed by the spores was determined by plating samples onto brain heart infusion agar, incubating anaerobically at 37 °C for 24 h, and counting the colonies formed. Black bars represent initial viable CFU counts; gray bars represent viable CFU counts after 6 h of anaerobic incubation at 37 °C for 24 h, and (–) represents negative control to verify that the meat samples were free of naturally occurring microorganisms. Error bars represent the standard deviation from the means of two independent experiments using two independent spore preparations.

**Table 1 microorganisms-13-00072-t001:** Antibacterial activity of raw natural substances against *Clostridium perfringens* SM101, determined by agar diffusion assay.

Raw Product (Powder or Juice)	Solvent	Inhibition Zone of *C. perfringens* SM101 (Mean ± SD; mm)
Onion juice (undiluted)	-	15.9 ± 1.13
Garlic juice (undiluted)	-	29.6 ± 3.16
Ginger juice (undiluted)	-	0 ± 0
Cinnamon powder (20% *w*/*v*)	SDW	0 ± 0
	20% DMSO	11.75 ± 0.35 *
Garlic powder (20% *w*/*v*)	SDW	24.5 ± 0.5
	20% DMSO	25.75 ± 4.5
Turmeric powder (20% *w*/*v*)	SDW	0 ± 0
	20% DMSO	9.1 ± 0.14 *
Ginger powder (20% *w*/*v*)	SDW	0 ± 0
	20% DMSO	0 ± 0

SD, standard deviation; DMSO, dimethyl sulfoxide; SDW, sterile distilled water. Negative controls included SDW and 20% DMSO (no zone of inhibition was observed with either). * *p* values indicate the difference in performance between each product diluted in the two solvents against strain SM101 (*p* < 0.05).

**Table 2 microorganisms-13-00072-t002:** Evaluation of the antibacterial activities of processed natural substances against *C. perfringens* SM101 using the method of agar diffusion assay.

Processed Products (Tablets, Capsules, and Oils)	Solvent	Inhibition Zone of *C. perfringens* SM101 (Mean ± SD; mm)
Garlic tablets (20% *w*/*v*)	SDW	26.5 ± 0.70
	20% DMSO	26.25 ± 1.7
Turmeric capsules (20% *w*/*v*)	SDW	0 ± 0
	20% DMSO	11 ± 1.4 *
Ginger tablets (20% *w*/*v*)	SDW	0 ± 0
	20% DMSO	0 ± 0
Cinnamon capsules 20% *w*/*v*)	SDW	0 ± 0
	20% DMSO	13.4 ± 0.14 *
Coconut oil	-	0 ± 0
Peppermint oil	50% coconut oil	21.3 ± 0.7
Clove oil 50%	50% coconut oil	16.75 ± 0.5
Rosemary oil 50%	50% coconut oil	15.75 ± 0.5

SD, standard deviation; DMSO, dimethyl sulfoxide; SDW, sterile distilled water. Negative controls included SDW and 20% DMSO (no zone of inhibition was observed with either). * *p* values indicate the difference in performance between each product diluted in the two solvents against strain SM101 (*p* < 0.05).

## Data Availability

The original contributions presented in this study are included in the article. Further inquiries can be directed to the corresponding author.

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
