# Peer review of "Inhibitory Effects of Natural Products on Germination, Outgrowth, and Vegetative Growth of Clostridium perfringens Spores in Laboratory Medium and Chicken Meat"

_microorganisms, 2025, doi:10.3390/microorganisms13010072_

Round 1
Reviewer 1 Report
Comments and Suggestions for Authors
Dear Authors,
The manuscript titled "The inhibitory effects of natural products on germination, outgrowth, and vegetative growth of Clostridium perfringens spores in laboratory medium and chicken meat" investigates the antimicrobial properties of 15 natural products. Garlic, onion juice, and essential oil constituents (EOCs) from clove, rosemary, and peppermint were identified as the most effective.
In laboratory media (TGY), clove and peppermint oils (0.5%) successfully inactivated spore germination, while EOCs (0.1–0.5%) inhibited spore outgrowth and completely suppressed vegetative growth at 0.5% during a 6-hour incubation. However, in contaminated chicken meat stored under abusive conditions, even a fourfold higher concentration (2%) of EOCs failed to inhibit spore growth.
These findings demonstrate the strong antimicrobial activity of clove and peppermint EOCs in controlled environments but underscore their limitations in real food systems, highlighting the challenges of applying natural antimicrobials in complex matrices.
During the review, I highlighted the following critical points
Introduction
Strengths:
- Provides a comprehensive background on Clostridium perfringens and its impact.
- Justifies the importance of natural antimicrobials in food safety.
Critical Issues:
- The introduction is overly detailed, with tangential information about alternative antimicrobial technologies that distracts from the main research focus.
- Limited explanation of why natural antimicrobials were chosen over synthetic alternatives in this study.
Suggestions:
- Streamline the discussion of alternatives to focus on their relevance to the study.
- Provide a clearer rationale for the focus on natural antimicrobials, emphasizing consumer demand or regulatory constraints.
Materials and Methods
Strengths:
- Detailed experimental setup ensures reproducibility.
- Methodological rigor is evident in the preparation of bacterial strains, spore assays, and statistical analysis.
Critical Issues:
- Overly technical descriptions in some sections may hinder readability for non-specialist readers.
- Lack of explanation for why certain natural products were prioritized for testing (e.g., historical use or scientific rationale).
Suggestions:
- Simplify language in the methodological descriptions without losing precision.
- Include a brief explanation for the selection of specific natural products.
Results and Discussion
Strengths:
- Results are well-supported by data, including detailed graphs and statistical analyses.
- Discussion highlights comparisons with previous studies.
Critical Issues:
- The laboratory findings are emphasized, but the discussion of real-world applications (e.g., chicken meat systems) is underdeveloped.
- The potential reasons for the reduced efficacy of EOCs in meat matrices are not thoroughly explored.
- Limited reflection on the practical implications or next steps for applying these findings.
Suggestions:
- Expand the discussion of why EOCs underperform in complex food systems, incorporating more hypotheses or previous research.
- Include a section on practical implications, such as potential formulation adjustments or combinations with other preservation strategies.
Conclusion
Strengths:
- Concisely summarizes findings.
Critical Issues:
- Overly general; it does not provide clear next steps or implications for food safety applications.
- The conclusion reiterates findings without discussing their broader significance or how limitations could be addressed.
Suggestions:
- Highlight specific next steps for research or practical application.
- Emphasize the study's contribution to the understanding of natural antimicrobials in food safety.
References
Strengths:
- Comprehensive and relevant references are included.
Author Response
Response to reviewer #1 comments is enclosed

Reviewer 2 Report
Comments and Suggestions for Authors
There are some points that should be taken into consideration while reviewing the manuscript:
l In the introduction part, it is necessary to highlight the rationale behind studying the inhibitory effect of these plants and essential oils in particular.
l In the Materials and Methods section, line 157, it is stated that “Fifteen natural products were purchased.” Please list these fifteen products.
l Line 268, it was mentioned that two-tailed student’s t-test was used in the statistical analysis. The two-tailed student’s t-test is a statistical test used to test whether the difference between the response of two groups while the comparison in the study is between more than two groups.
l The discussion part requires further interpretation of the results in light of the bioactive compounds of the natural products studied, as well as comparison with the results of previous studies on the same natural products, even if they were on other strains of Clostridium or other pathogenic bacteria.
l The conclusion part needs to be re-edited and clarified which treatments were most promising and effective and what the final recommendation of the study is based on the results.

Author Response
Response to Reviewer #2 comments is enclosed

Round 2
Reviewer 2 Report
Comments and Suggestions for Authors
Many thanks to the authors for answering the queries and making the suggested modifications.